# SpikeBERT: A Language Spikformer Trained with Two-Stage Knowledge Distillation from BERT

## Abstract

Spiking neural networks (SNNs) offer a promising avenue to implement deep neural networks in a more energy-efficient way. However, the network architectures of existing SNNs for language tasks are too simplistic, and deep architectures have not been fully explored, resulting in a significant performance gap compared to mainstream transformer-based networks such as BERT. To this end, we improve a recently-proposed spiking transformer (i.e., Spikformer) to make it possible to process language tasks and propose a two-stage knowledge distillation method for training it, which combines pre-training by distilling knowledge from BERT with a large collection of unlabelled texts and fine-tuning with task-specific instances via knowledge distillation again from the BERT fine-tuned on the same training examples. Through extensive experimentation, we show that the models trained with our method, named SpikeBERT, outperform state-of-the-art SNNs and even achieve comparable results to BERTs on text classification tasks for both English and Chinese with much less energy consumption.

## 1 Introduction

Modern artificial neural networks (ANNs) have been highly successful for a wide range of natural language processing (NLP) and computer vision (CV) tasks. However, it requires too much computational power and energy to train and deploy state-of-the-art ANN models, leading to a consistent increase of energy consumption per model over the past decade. The energy consumption of large language models, such as ChatGPT[OpenAI, 2022] and GPT-4[OpenAI, 2023], is unfathomable even during inference. In recent years, spiking neural networks (SNNs), arguably known as the third generation of neural network [Maas, 1997], have attracted a lot of attention due to their high biological plausibility, event-driven property and low energy consumption [Roy et al., 2019]. Like biological neurons, SNNs use discrete spikes to process and transmit information. Nowadays, neuromorphic hardware can be used to fulfill spike-based computing, which provides a promising way to implement artificial intelligence with much lower energy consumption.

Spiking neural networks have achieved great success in image classification task [Hu et al., 2018, Yin et al., 2020, Fang et al., 2021, Ding et al., 2021, Kim et al., 2022a, Zhou et al., 2022] and there have been some works [Plank et al., 2021, Lv et al., 2023, Zhu et al., 2023] that have demonstrated the efficacy of SNNs in language tasks partially. However, the backbone networks employed in SNNs for language tasks are overly simplistic, which significantly lowers the upper bound on the performance of their SNN models. For instance, the SNN used by Lv et al. [2023], which is built upon TextCNN [Kim, 2014], demonstrates a notable performance gap compared to those built on Transformer-based

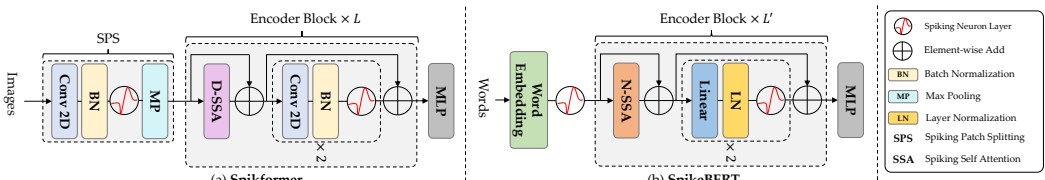

Figure 1: (a) The architecture of Spikformer [Zhou et al., 2022]. (b) The architecture of our SpikeBERT. We improve Spikformer in its architecture, making it possible to process languages. Firstly, Spiking Patch Splitting (SPS) module was replaced with a word embedding layer so that the network can take discrete words (or tokens) as input. Secondly, we make the shape of the attention map yielded by Spiking Self Attention (SSA) to be $N \times N$, rather than $D \times D$, where $D$ and $N$ denote dimensionality of hidden layers and the length of inputs respectively. Lastly, the convolution layers were replaced with linear layers, and the batch normalization with layer normalization. $L$ and $L^{'}$ denote the number of encoder blocks in Spikfomer and SpikeBERT, respectively.

[Vaswani et al., 2017] large language models like BERT [Devlin et al., 2019] and RoBERTa [Liu et al., 2019] on multiple classification benchmarks.

Recently, Spikformer was proposed by Zhou et al. [2022], which first introduced Transformer architecture to SNNs and significantly narrowed the gap between SNNs and ViT [Dosovitskiy et al., 2020] on ImageNet [Deng et al., 2009] and CIFAR-10. We think that Spikformer provides the possibility to construct complex language representation models. As shown in Figure 1, considering the discrete nature of textual data, we improve the architecture of Spikformer to make it suitable for language tasks. we replace certain modules that were originally designed for image processing with language-friendly modules. Please see Section 3.2 for details on the improvement in network architecture. In general, a deeper ANN model often implies better performance. Increasing the depth of a ANN allows for the extraction of more complex and abstract features from the input data. However, Fang et al. [2020a] have shown that deep SNNs directly trained with backpropagation through time (BPTT) [Werbos, 1990] using surrogate gradients (See Section2.1) could suffer from the problem of gradient vanishing or exploding due to "self-accumulating dynamics". Therefore, we proposed to use knowledge distillation [Hinton et al., 2015] to train language Spikformers so that the deviation of surrogate gradients in Spikformer would not be rapidly accumulated[Qiu et al., 2023].

Inspired by the widely-used "pre-training + fine-tuning" recipe [Sun et al., 2019, Liu, 2019, Gururangan et al., 2020], we present a two-stage knowledge distillation strategy. In stage 1, we choose BERT as teacher model and the improved Spikformer as student model. We utilize a large collection of unlabelled texts to align features produced by two models in the embedding layer and multiple hidden layers. In stage 2, we use a BERT fine-tuned on a task-specific dataset as teacher and the model that completes stage 1 as student. At this stage, we first do the data augmentation for task-specific dataset and then employ the logits predicted by the teacher model to further guide the student model. After two-stage knowledge distillation, a spiking language model, named SpikeBERT, can be built by distilling knowledge from BERT. The experiment results show that SpikeBERT not only can outperform the state-of-the-art SNNs-like frameworks in text classification task but also achieve competitive performance to BERTs. The experiments of the ablation study (Section 4.5) also show that "pre-training distillation" plays an important role in training SpikeBERT.

The major contribution of this study can be summarized as follows:

- We improve the architecture of Spikformer for language processing and propose a two-stage, "pre-training + task-specific" knowledge distillation training method, in which the improved Spikformers are pre-trained on a huge collection of unlabelled texts before they are further fine-tuned on task-specific datasets by distilling the knowledge of feature extractions and predictive powers from BERTs.
- We empirically show that SpikeBERT achieved significantly higher performance than existing SNNs on 6 different language benchmark datasets for both English and Chinese.

- This study is among the first to show the feasibility of transferring the knowledge of BERT-like large language models to spiking-based architectures that can achieve comparable results but with much less energy consumption.

# 2 Related Work

## 2.1 Spiking Neural Networks

SNNs use discrete spike trains instead of continuous decimal values to compute and transmit information. Spiking neurons, such as Izhikevich neuron [Izhikevich, 2003] and Leaky Integrate-and-Fire (LIF) neuron [Wu et al., 2017], are usually applied to generate spike trains. However, due to the non-differentiability of spikes, training SNNs has been a great challenge for the past two decades. Currently, there are two mainstream approaches to address this problem.

**ANN-to-SNN Conversion** ANN-to-SNN conversion method [Diehl et al., 2015, Cao et al., 2015, Rueckauer et al., 2017, Hu et al., 2018] aims to convert weights of a well-trained ANN to its SNN counterpart by replacing the activation function with spiking neuron layers and adding scaling rules such as weight normalization [Diehl et al., 2016] and threshold constraints [Hu et al., 2018]. This approach suffers from a large number of time steps during the conversion.

**Backpropagation with Surrogate Gradients** Another popular approach is to utilize surrogate gradients [Neftci et al., 2019] during error backpropagation, enabling the entire procedure to be differentiable. Multiple surrogate gradients functions have been proposed, including the Sigmoid surrogate function [Zenke and Ganguli, 2017], Fast-Sigmoid [Zheng and Mazumder, 2018], and ATan [Fang et al., 2020a]. Backpropagation through time (BPTT) [Werbos, 1990] is one of the most popular methods for directly training SNNs[Shrestha and Orchard, 2018, Kang et al., 2022], which applies the traditional backpropagation algorithm [LeCun et al., 1989] to the unrolled computational graph. In recent years, several BPTT-like training strategies have been proposed, including SpatioTemporal Backpropagation (STBP) [Wu et al., 2017], STBP with Temporal Dependent Batch Normalization (STBP-tdBN) [Zheng et al., 2020], and Spatio-Temporal Dropout Backpropagation (STDB) [Rathi et al., 2020]. These strategies have demonstrated high performance under specific settings. For more detailed information about Backpropagation Through Time (BPTT), please refer to Appendix A.

## 2.2 Knowledge Distillation

Hinton et al. [2015] proposed the concept of knowledge distillation by utilizing the "response-based" knowledge (i.e., soft labels) of the teacher model to transfer knowledge. However, when this concept was first proposed, the features captured in the hidden layers were neglected, as they only focused on the final probability distribution at that time. To better learn from teacher models, some works [Zagoruyko and Komodakis, 2016, Heo et al., 2019, Chen et al., 2021] have advocated for incorporating hidden feature alignment during the distillation process. In addition, relation-based knowledge distillation has been introduced by Park et al. [2019], demonstrating that the interrelations between training data examples were also essential.

Recently, there have been a few studies [Kushawaha et al., 2020, Takuya et al., 2021, Qiu et al., 2023] in which knowledge distillation approaches were introduced to train SNNs. However, most of them focused on image classification task only, which cannot be trivially applied to language tasks. In this study, we propose a two-stage knowledge distillation approach to train the proposed SpikeBERT for text classification tasks, which is among the first ones to show the feasibility of transferring the knowledge to SNNs from large language models.

# 3 Method

In this section, we describe how we improve the architecture of Spikformer and introduce our two-stage distillation approach for training SpikeBERT. Firstly, we will depict how spiking neurons and surrogate gradients work in spiking neural networks. Then we will show the simple but effective

modification of Spikformer to enable it to represent text information. Lastly, we will illustrate "pre-training + task-specific" distillation in detail.

## 3.1 Spiking Neurons and Surrogate Gradients

Leaky integrate-and-fire (LIF) neuron [Wu et al., 2017] is one of the most widely used spiking neurons. Similar to the traditional activation function such as ReLU, LIF neurons operate on a weighted sum of inputs, which contributes to the membrane potential $U_t$ of the neuron at time step $t$. If membrane potential of the neuron reaches a threshold $U_{\text{thr}}$, a spike $S_t$ will be generated:

$$S_t = \begin{cases} 1, & \text{if } U_t \geq U_{\text{thr}}; \\ 0, & \text{if } U_t < U_{\text{thr}}. \end{cases} \tag{1}$$

We can regard the dynamics of the neuron's membrane potential as a resistor-capacitor circuit [Maas, 1997]. The approximate solution to the differential equation of this circuit can be represented as follows:

$$U_t = I_t + \beta U_{t-1} - S_{t-1} U_{\text{thr}}, \quad I_t = W X_t \tag{2}$$

where $X_t$ are inputs to the LIF neuron at time step $t$, $W$ is a set of learnable weights used to integrate different inputs, $I_t$ is the weighted sum of inputs, $\beta$ is the decay rate of membrane potential, and $U_{t-1}$ is the membrane potential at time $t-1$. The last term of $S_{t-1} U_{\text{thr}}$ is introduced to model the spiking and membrane potential reset mechanism.

In addition, we follow Fang et al. [2020b] and use Arctangent-like surrogate gradients function, which regards the Heaviside step function (Equation 1) as:

$$S \approx \frac{1}{\pi} \arctan(\frac{\pi}{2} \alpha U) + \frac{1}{2} \tag{3}$$

Therefore, the gradients of $S$ in Equation 3 are:

$$\frac{\partial S}{\partial U} = \frac{\alpha}{2} \frac{1}{(1 + (\frac{\pi}{2} \alpha U)^2)} \tag{4}$$

where $\alpha$ defaults to 2.

## 3.2 SpikeBERT Architecture

Spikformer [Zhou et al., 2022] is the first hardware-friendly Transformer-based spiking neural network, whose architecture is shown in Figure 1 (a). The most crucial module is the Spiking Self Attention (SSA), which utilizes discrete spikes to implement the self-attention mechanism without employing a softmax function:

$$\text{SSA}(Q_s, K_s, V_s) = \text{S(BN(MLP}(Q_s K_s^T V_s * \tau)))$$
$$Q_s = \text{S}_{\text{Q}_s}(\text{BN}(X_s W_{Q_s})), \quad K_s = \text{S}_{\text{K}_s}(\text{BN}(X_s W_{K_s})), \quad V_s = \text{S}_{\text{V}_s}(\text{BN}(X_s W_{V_s})) \tag{5}$$

where S is Heaviside step function like Equation 1, $X_s \in \mathbb{R}^{T \times L \times D}$ is the input of SSA, $T$ is number of time steps, BN is batch normalization, $\tau$ is a scaling factor. Outputs of SSA and $Q_s, K_s, V_s$ are all matrix containing 0 and 1. $W_{Q_s}, W_{K_s}, W_{V_s}$ and MLP are all learnable decimal parameters.

We modify Spikformer so that it can effectively process textual data. Firstly, we replace Spiking Patch Splitting (SPS) module with a word embedding layer and a spiking neuron layer so that it can process sentences. Meanwhile, we find that the shape of the attention map in vanilla Spikformer is $D \times D$ where $D$ is the dimensionality of the hidden layers, which is unreasonable in language tasks. For language tasks, the features shared with words in different positions by attention mechanism are more important than those in different dimensions. Therefore, we reshape the attention map in Spiking Self Attention (SSA) module to $N * N$ where $N$ is the length of inputs. Lastly, we use linear layers and layer normalization (LN) instead of convolution layers and batch normalization(BN). We show the architecture of SpikeBERT in Figure 1 (b).

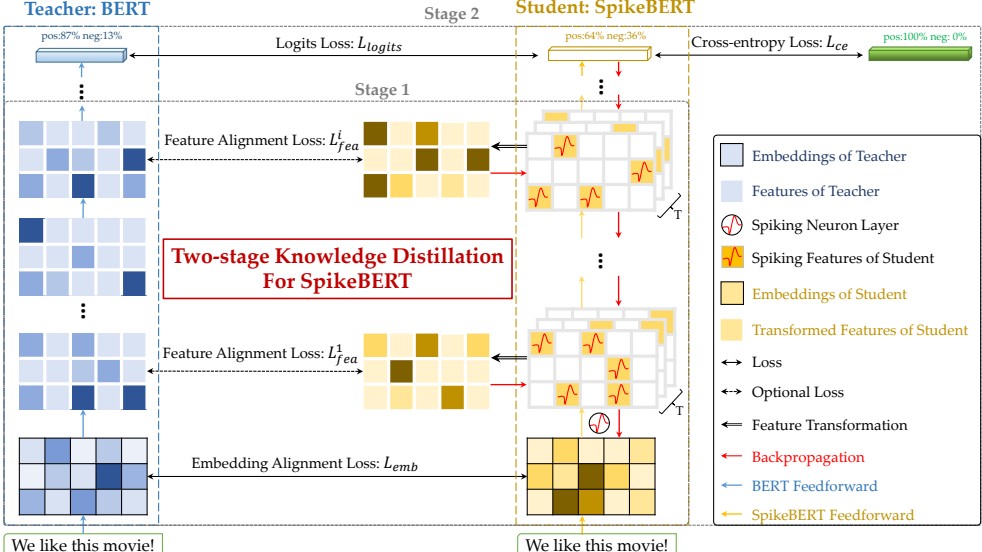

Figure 2: Overview of our two-stage distillation method (pre-training + task-specific distillation) for training SpikeBERT. $T$ is the number of time steps of features in every layer. Notice that the logits loss and cross-entropy loss are only considered in stage 2. The varying shades of color represent the magnitude of the floating-point values. The dotted line under $L^i_{fea}$ indicates that features of some hidden layers can be ignored when calculating feature alignment loss. If the student model contains different numbers of layers from the teacher model, we will align features every few layers.

### 3.3 Two-stage Distillation

Two-stage distillation is the key to enabling the student model with language processing ability. The first stage is to align the embeddings and hidden features between BERT and the improved Spikformer using a large-scale corpus. The second stage is to distill logits and cross-entropy information on a task-specific dataset from a fine-tuned BERT to the model finishing stage 1. We show the overview of our method in Figure 2.

#### 3.3.1 Stage 1. Pre-training Distillation

Given a pre-trained BERT [Devlin et al., 2019] irrelevant to downstream tasks as teacher $TM$ and an improved Spikformer as student $SM$, our goal in this stage is to align the embeddings and hidden features of $TM$ and $SM$ with a collection of unlabelled texts. We will introduce embedding alignment loss and feature alignment loss in the following.

**Feature Alignment Loss** This loss $L_{fea}$ is to measure the similarity of features between $TM$ and $SM$ at every hidden layer. However, the shape of the student model's feature $F_{sm}$ at every layer is $T \times N \times D$ but that of BERT's feature $F_{tm}$ is $N \times D$, where $T$ is the number of time steps, $D$ is the dimensionality of hidden layers and $L$ is sample length. What's more, $F_{sm}$ is a matrix only containing 0 and 1 but $F_{tm}$ is a decimal matrix. To address the issue of different dimensions between $F_{tm}$ and $F_{sm}$, as well as the disparity between continuous features of $TM$ and discrete features of $SM$, a transformation strategy is necessary. We follow the feature transformation approaches of Heo et al. [2019], Chen et al. [2021], Qiu et al. [2023] to map the features of $TM$ and $SM$ to the same content space:

$$F'_{tm} = F_{tm}, \quad F'_{sm} = \text{LayerNorm}(\text{MLP}(\sum_{t}^{T}(F^t_{sm}))) \quad (6)$$

However, we find it hard to align the features generated by the student model with those generated by BERT for the first few layers in this stage. We think that's because the student model might require more network layers to capture the essential features via the interaction among the inputs. As shown in Figure 2, we choose to ignore some front layers when calculating feature alignment loss. Assume

BERT contains $B$ Transformer blocks (i.e., $B$ layers) and assume the student model contains $M$ Spike Transformer Block. Therefore, we will align features every $\lceil \frac{B}{M} \rceil$ layers if $B > M$. For layer $i$ in student model, its feature alignment loss is $L_{fea}^i = ||F_{tm}^{'} - F_{sm}^{'}||_2$.

**Embedding Alignment Loss**    As discussed in Section 3.2, the embeddings of the input sentences are not in the form of spikes until they are fed forward into the Heaviside step function. Define $E_{tm}$ and $E_{sm}$ as the embeddings of teacher and student, respectively so the feature alignment loss is $L_{fea}^i = ||E_{tm} - \mathrm{MLP}(E_{sm})||_2$. The MLP layer is a transformation playing a similar role as that in Equation 6.

To sum up, in stage 1, the total loss $L_1$ is the sum of chosen layer's feature alignment loss:

$$L_1 = \sigma_1 \sum_i L_{fea}^i + \sigma_2 L_{emb} \tag{7}$$

where the hyperparameters $\sigma_1$ and $\sigma_2$ are used to balance the learning of embeddings and features.

### 3.3.2   Stage 2. Task-specific Distillation

In stage 2, we take a BERT fine-tuned on a task-specific dataset as the teacher model, and the model completed stage 1 as the student. To accomplish a certain language task, there should be a task-specific head over the basic language model as shown in Figure 2. For example, it is necessary to add an MLP layer over BERT for text classification. Besides, data augmentation is a commonly used and highly effective technique in knowledge distillation[Jiao et al., 2019, Tang et al., 2019, Liu et al., 2022]. In the following, we will discuss our approach to data augmentation, as well as the logits loss and cross-entropy loss.

**Data Augmentation**    In the distillation approach, a small dataset may be insufficient for the teacher model to fully express its knowledge[Ba and Caruana, 2013]. To tackle this issue, we augment the training set in order to facilitate effective knowledge distillation. We follow Tang et al. [2019] to augment the training set:

- Firstly, we randomly replace a word with [MASK] token with probability $p_{mask}$.
- Secondly, we replace a word with another of the same POS tag with probability $p_{pos}$.
- Thirdly, we randomly sample an $n$-gram from a training example with probability $p_{ng}$, where $n$ is randomly selected from $\{1, 2, ..., 5\}$.

**Logits Loss**    Following Hinton et al. [2015], we take logits, also known as soft labels, into consideration, which lets the student learn the prediction distribution of the teacher. To measure the distance between two distributions, we choose KL-divergence: $L_{logits} = \sum_i^c p_i log\left(\frac{p_i}{q_i}\right)$, where $c$ is the number of categories, $p_i$ and $q_i$ denote the prediction distribution of the teacher model and student model.

**Cross-entropy Loss**    Cross-entropy loss can help the student model learn from the samples in task-specific datasets: $L_{ce} = -\sum_i^c \hat{q}_i log\left(q_i\right)$, where $\hat{q}_i$ represents the one-hot label vector.

Therefore, the total loss $L_2$ of stage 2 contains three terms:

$$L_2 = \lambda_1 \sum_i L_{fea}^i + \lambda_2 L_{emb} + \lambda_3 L_{logits} + \lambda_4 L_{ce} \tag{8}$$

where $\lambda_1$, $\lambda_2$, $\lambda_3$, and $\lambda_4$ are the hype-parameters that control the weight of these loss.

For both stages, we adopt backpropagation through time (BPTT), which is suitable for training spiking neural networks. You can see the detailed derivation in Appendix A if interested.

## 4   Experiments

We conduct four sets of experiments. The first is to evaluate the accuracy of SpikeBERT trained with the proposed method on 6 datasets of text classification datasets. The second experiment is to compare the theoretical energy consumption of BERT and that of SpikeBERT. The third experiment is an ablation study about the training process. The last experiment is to figure out how the performance of SpikeBERT is impacted by the number of time steps and model depth.

## 4.1 Datasets

As mentioned in Section 3.3.1, a large-scale parallel corpus will be used to train student models in Stage 1. For the English corpus, we choose the "20220301.en" subset of Wikipedia[1] and the whole Bookcorpus[Zhu et al., 2015], which are both utilized to pre-train a BERT [Devlin et al., 2019]. For the Chinese corpus, we choose Chinese-Wikipedia dump[2] (as of Jan. 4, 2023). Additionally, we follow Lv et al. [2023] to evaluate the SpikeBERT trained with the proposed distillation method on six text classification datasets: MR[Pang and Lee, 2005], SST-2[Socher et al., 2013], SST-5, Subj, ChnSenti, and Waimai. The dataset details are provided in Appendix B.

## 4.2 Implementation Details

Firstly, we set the number of encoder blocks in SpikeBERT to 12. Additionally, we set the threshold of common spiking neurons $U_{thr}$ as 1.0 but set the threshold of neurons in the spiking self-attention block as 0.25 in SpikeBERT. In addition, we set decay rate $\beta = 0.9$ and scaling factor $\tau$ as 0.125. We also set the time step $T$ of spiking inputs as 4 and sentence length to 256 for all datasets.

To construct SpikeBERT, we use two Pytorch-based frameworks: SnnTorch [Eshraghian et al., 2021] and SpikingJelly [Fang et al., 2020b]. Besides, we utilize bert-base-cased [3] from Huggingface as teacher model for English datasets and Chinese-bert-wwm-base[4] [Cui et al., 2019] for Chinese datasets.

In addition, we conduct pre-training distillation on 4 NVIDIA A100-PCIE GPUs and task-specific distillation on 4 NVIDIA GeForce RTX 3090 GPUs. Since surrogate gradients are required during backpropagation, we set $\alpha$ in Equation 3 as 2. In stage 1, we set the batch size as 128 and adopt AdamW [Loshchilov and Hutter, 2017] optimizer with a learning rate of $5e^{-4}$ and a weight decay rate of $5e^{-3}$. The hyperparameters $\sigma_1$ and $\sigma_2$ in Equation 7 are both set to 1.0. In stage 2, we set the batch size as 32 and the learning rate to $5e^{-5}$. For data augmentation, we set $p_{mask} = p_{pos} = 0.1$, $p_{ng} = 0.25$. To balance the weights of the four types of loss in Equation 8, we set $\lambda_1 = 0.1$, $\lambda_2 = 0.1$, $\lambda_3 = 1.0$, and $\lambda_4 = 0.1$.

## 4.3 Results

We report in Table 1 the accuracy achieved by SpikeBERT trained with "pre-training + task-specific" distillation on 6 datasets, compared to 2 baselines: 1) SNN-TextCNN proposed by Lv et al. [2023]; 2) improved Spikformer directly trained with gradient descent algorithm using surrogate gradients.

Table 1: Classification accuracy achieved by different methods on 6 datasets. A BERT model fine-tuned on the dataset is denoted as "FT BERT". The improved Spikformer directly trained with surrogate gradients on the dataset is denoted as "Directly-trained Spikformer". All reported experimental results are averaged across 10 random seeds.

| Model | English Dataset | | | | Chinese Dataset | | Avg. |
|---|---|---|---|---|---|---|---|
| | MR | SST-2 | Subj | SST-5 | ChnSenti | Waimai | |
| TextCNN [Kim, 2014] | 77.41 | 83.25 | 94.00 | 45.48 | 86.74 | 88.49 | 79.23 |
| FT BERT [Devlin et al., 2019] | 87.63 | 92.31 | 95.90 | 50.41 | 89.48 | 90.27 | 84.33 |
| SNN-TextCNN [Lv et al., 2023] | 75.45 | 80.91 | 90.60 | 41.63 | 85.02 | 86.66 | 76.71 |
| Directly-trained Spikformer | 76.38 | 81.55 | 91.80 | 42.02 | 85.45 | 86.93 | 77.36 |
| SpikeBERT [Ours] | **80.69** | **85.39** | **93.00** | **46.11** | **86.36** | **89.66** | **80.20** |

Table 1 demonstrates that the SpikeBERT trained with two-stage distillation achieves state-out-of-art performance across 6 text classification datasets. Compared to SNN-TextCNN, SpikeBERT achieved up to $5.42\%$ improvement in accuracy ($3.49\%$ increase on average) for all text classification benchmarks. Furthermore, SpikeBERT outperforms TextCNN, which is considered a representative

---

[1] https://dumps.wikimedia.org/

[2] https://dumps.wikimedia.org/zhwiki/latest/

[3] https://huggingface.co/bert-base-cased

[4] https://huggingface.co/hfl/chinese-bert-wwm

artificial neural network, and even achieves comparable results to the fine-tuned BERT by a small drop of $4.13\%$ on average in accuracy for text classification task. What's more, Table 1 demonstrates that SpikeBERT can also be applied well in Chinese datasets (ChnSenti and Waimai).

Fang et al. [2020a] propose that, in image classification task, surrogate gradients of SNNs may lead to gradient vanishing or exploding and it is even getting worse with the increase of model depth. We found this phenomenon in language tasks as well. Table 1 reveals that the accuracy of directly-trained Spikformer is noticeably lower than SpikeBERT on some benchmarks, such as MR, SST-5, and ChnSenti. This is likely because the directly-trained Spikformer models have not yet fully converged due to gradient vanishing or exploding.

## 4.4 Energy Consumption

An essential advantage of SNNs is the low consumption of energy during inference. We compare the theoretical energy consumption per sample of fine-tuned BERT and SpikeBERT on 6 test datasets and report the results in Table 2. The way to calculate floating point operations (FLOPs), synaptic operations (SOPs), and the theoretical energy consumption (Power) is shown in Appendix C.

Table 2: Energy consumption per sample of fine-tuned BERT and SpikeBERT during inference on 6 text classification benchmarks. "FLOPs" denotes the floating point operations of fine-tuned BERT. "SOPs" denotes the synaptic operations of SpikeBERT. "Power" denotes the average theoretical energy required for each test example prediction.

| Dataset | Model | FLOPs / SOPs(G) | Power (mJ) | Energy Reduction | Accuracy (%) |
|---|---|---|---|---|---|
| ChnSenti | FT BERT | 22.46 | 103.38 | **73.28**% ↓ | 89.48 |
| | SpikeBERT | 28.47 | 27.62 | | 86.36 |
| Waimai | FT BERT | 22.46 | 103.38 | **73.91**% ↓ | 90.27 |
| | SpikeBERT | 27.81 | 26.97 | | 89.66 |
| MR | FT BERT | 22.23 | 102.24 | **74.93**% ↓ | 87.63 |
| | SpikeBERT | 26.94 | 25.63 | | 80.69 |
| SST-2 | FT BERT | 22.23 | 102.24 | **73.78**% ↓ | 92.31 |
| | SpikeBERT | 27.46 | 26.81 | | 85.39 |
| Subj | FT BERT | 22.23 | 102.24 | **77.17**% ↓ | 95.90 |
| | SpikeBERT | 25.92 | 23.34 | | 93.00 |
| SST-5 | FT BERT | 22.23 | 102.24 | **76.92**% ↓ | 50.41 |
| | SpikeBERT | 26.01 | 23.60 | | 46.11 |

It is worth noting that the energy consumption of SpikeBERT is significantly lower than that of fine-tuned BERT, which is an important advantage of SNNs over ANNs in terms of energy efficiency. As shown in Table 2, SpikeBERT demands only $25.00\%$ of the energy that fine-tuned BERT needs to achieve comparable performance on average. Moreover, on the Subj dataset, SpikeBERT can reduce energy consumption by up to $77.17\%$ compared to fine-tuned BERT for predicting each text example. This indicates that SpikeBERT is a promising candidate for energy-efficient text classification in resource-constrained scenarios.

## 4.5 Ablation Study and Impact of Hyper-parameters

In this section, we conduct ablation studies to investigate the contributions of: a) different stages of the proposed knowledge distillation method, and b) different types of loss in Equation 8.

As we can see in Table 4.5, SpikeBERTs without either stage 1 or stage 2 experience about $3.20\%$ performance drop on average. Therefore, we conclude that the two distillation stages are both essential for training SpikeBERT. Furthermore, we observed that the average performance dropped from 76.30 to 73.27 when excluding the logits loss, demonstrating that the logits loss $L_{logits}$ has the greatest impact on task-specific distillation. Meanwhile, data augmentation (DA) plays an important role in Stage 2, contributing to an increase in average performance from 75.54 to 76.30.

We investigate how the performance of SpikeBERT is affected by the two important hyperparameters: time steps $T$ and model depth. To this end, we conduct two experiments: (a) varying the number of

Table 3: Ablation studies of the two-stage distillation method. Row 3 and 4 show ablation experiment results on the two steps of our proposed method. Row 5 to 9 are ablation experiment results on different parts of Equation 8. "DA" stands for data augmentation.

| | Models | MR | SST-2 | Subj | SST-5 | Avg. | Drop |
|---|---|---|---|---|---|---|---|
| | SpikeBERT | 80.69 | 85.39 | 93.00 | 46.11 | 76.30 | – |
| | w/o Stage 1 | 76.04 | 82.26 | 91.80 | 42.16 | 73.07 | -3.23 |
| | w/o Stage 2 | 75.91 | 82.26 | 91.90 | 42.58 | 73.14 | -3.16 |
| Stage 2 | w/o DA | 80.22 | 84.90 | 92.20 | 44.84 | 75.54 | -0.76 |
| | w/o $L_{fea}$ | 78.35 | 83.48 | 92.20 | 43.57 | 74.40 | -1.90 |
| | w/o $L_{emb}$ | 79.67 | 83.10 | 92.00 | 43.48 | 74.56 | -1.74 |
| | w/o $L_{logits}$ | 76.19 | 82.64 | 91.90 | 42.35 | 73.27 | -3.03 |
| | w/o $L_{ce}$ | 80.43 | 85.23 | 93.00 | 45.86 | 76.13 | -0.17 |

the time steps of spike inputs when training SpikeBERT; and (b) training a variant of SpikeBERT with different encoder block depths, specifically $6, 12, 18$, using our proposed two-stage method.

Figure 3 (a) shows how the accuracy of SpikeBERT varies with the increase of time steps. We find that, with the increase of time steps, the accuracy increases first, then remains unchanged, and reaches its maximum roughly at $T = 4$. Theoretically, the performance of SpikeBERT should be higher with bigger time steps. However, the performance of models with $8$ and $12$ time steps is even worse than that with $4$ time steps on ChnSenti and Waimai datasets. A plausible explanation is that using excessively large time steps may introduce too much noise in the spike trains.

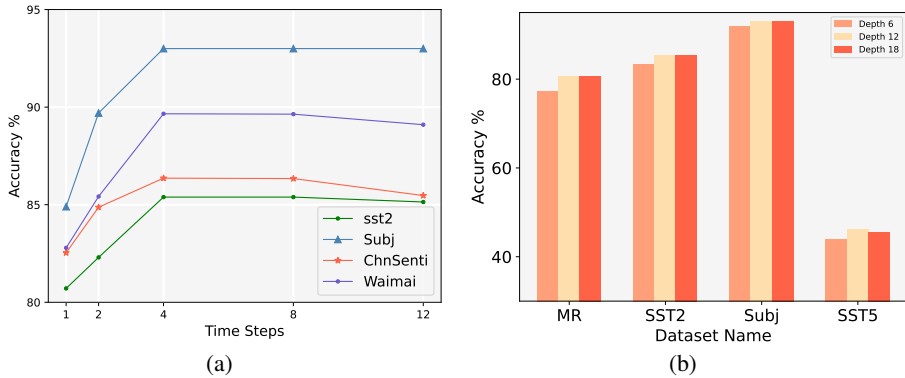

Figure 3: (a) Accuracy versus the number of time steps. (b) Accuracy influenced by model depth.

In addition, as we can see from Figure 3 (b), the accuracy of SpikeBERT is generally insensitive to the model depths and even gets lower in some datasets. We think that's because more spike Transformer blocks mean more spiking neurons (See Section 2.1), introducing more surrogate gradients when error backpropagation through time. Higher model depth often brings better model performance for traditional deep neural networks. However, it seems that deeper spiking neural networks cannot make further progress in performance. Many previous SNNs works [Zheng et al., 2020, Fang et al., 2020a, Kim et al., 2022b] have proved this deduction.

# 5   Conclusion

In this study, we extended and improved Spikformer to process language tasks and proposed a new promising training paradigm for training SpikeBERT inspired by the notion of knowledge distillation. We presented a two-stage, "pre-training + task-specific" knowledge distillation method by transferring the knowledge from BERTs to SpikeBERT for text classification tasks. We empirically show that our SpikeBERT outperforms the state-of-the-art SNNs and can even achieve comparable results to BERTs with much less energy consumption across multiple datasets for both English and Chinese, leading to future energy-efficient implementations of BERTs or large language models.

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

 **Appendix**

 ## A  Backpropagation Through Time in Spiking Neural Networks

452  The content of this section is mostly referred to Lv et al. [2023].

453  Given a loss function $L$ like Eqution 7 and 8, the losses at every time step can be summed together to
454  give the following global gradient:

$$\frac{\partial L}{\partial W} = \sum_t \frac{\partial L_t}{\partial W} = \sum_i \sum_{j \leq i} \frac{\partial L_i}{\partial W_j} \frac{\partial W_j}{\partial W} \tag{9}$$

455  where $i$ and $j$ denote different time steps, and $L_t$ is the loss calculated at time step $t$. No matter which
456  time step is, the weights of an SNN are shared across all steps. Therefore, we have $W_0 = W_1 =$
457  $\cdots = W$, which also indicates that $\frac{\partial W_j}{\partial W} = 1$. Thus, Equation (9) can be written as follows:

$$\frac{\partial L}{\partial W} = \sum_i \sum_{j \leq i} \frac{\partial L_i}{\partial W_j} \tag{10}$$

458  Based on the chain rule of derivatives, we obtain:

$$\begin{aligned}\frac{\partial L}{\partial W} &= \sum_i \sum_{j \leq i} \frac{\partial L_i}{\partial S_i} \frac{\partial S_i}{\partial U_i} \frac{\partial U_i}{\partial W_j} \\ &= \sum_i \frac{\partial L_i}{\partial S_i} \frac{\partial S_i}{\partial U_i} \sum_{j \leq i} \frac{\partial U_i}{\partial W_j}\end{aligned} \tag{11}$$

459  where $\frac{\partial L_i}{\partial S_i}$ is the derivative of the cross-entropy loss at the time step $i$ with respect to $S_i$, and $\frac{\partial S_i}{\partial U_i}$ can
460  be easily derived using surrogate gradients like Equation 3. As to the last term of $\sum_{j \leq i} \frac{\partial U_i}{\partial W_j}$, we can
461  split it into two parts:

$$\sum_{j \leq i} \frac{\partial U_i}{\partial W_j} = \frac{\partial U_i}{\partial W_i} + \sum_{j \leq i-1} \frac{\partial U_i}{\partial W_j} \tag{12}$$

462  From Equation (2), we know that $\frac{\partial U_i}{\partial W_i} = X_i$. Therefore, Equation (9) can be simplified as follows:

$$\frac{\partial L}{\partial W} = \sum_i \underbrace{\frac{\partial L_i}{\partial S_i} \frac{\partial S_i}{\partial U_i}}_{\text{constant}} \left( \underbrace{\frac{\partial U_i}{\partial W_j}}_{\text{constant}} + \sum_{j \leq i-1} \frac{\partial U_i}{\partial W_j} \right) \tag{13}$$

463  By the chain rule of derivatives over time, $\frac{\partial U_i}{\partial W_j}$ can be factorized into two parts:

$$\frac{\partial U_i}{\partial W_j} = \frac{\partial U_i}{\partial U_{i-1}} \frac{\partial U_{i-1}}{\partial W_j} \tag{14}$$

464  It is easy to see that $\frac{\partial U_i}{\partial U_{i-1}}$ is equal to $\beta$ from Equation (2), and Equation (9) can be written as:

$$\frac{\partial L}{\partial W} = \sum_i \underbrace{\frac{\partial L_i}{\partial S_i} \frac{\partial S_i}{\partial U_i}}_{\text{constant}} \left( \underbrace{\frac{\partial U_i}{\partial W_j}}_{\text{constant}} + \sum_{j \leq i-1} \underbrace{\frac{\partial U_i}{\partial U_{i-1}}}_{\text{constant}} \frac{\partial U_{i-1}}{\partial W_j} \right) \tag{15}$$

465  We can treat $\frac{\partial U_{i-1}}{\partial W_j}$ recurrently as Equation (12). Finally, we can update the weights $W$ by the rule of
466  $W = W - \eta \frac{\partial L}{\partial W}$, where $\eta$ is a learning rate.

 ## B  Datasets

468  The benchmark we used in Table 1 includes the following datasets:

- **MR**: MR stands for Movie Review and it consists of movie-review documents labeled with respect to their overall sentiment polarity (positive or negative) or subjective rating [Pang and Lee, 2005].
- **SST-5**: SST-5 contains $11,855$ sentences extracted from movie reviews for sentiment classification [Socher et al., 2013]. There are $5$ categories (very negative, negative, neutral, positive, and very positive).
- **SST-2**: The binary version of SST-5. There are just $2$ classes (positive and negative).
- **Subj**: The task of this dataset is to classify a sentence as being subjective or objective[5].
- **ChnSenti**: ChnSenti comprises about $7,000$ Chinese hotel reviews annotated with positive or negative labels[6].
- **Waimai**: There are about $12,000$ Chinese user reviews collected by a food delivery platform for binary sentiment classification (positive and negative)[7] in this dataset.

## C  Theoretical Energy Consumption Calculation

For spiking neural networks (SNNs), the theoretical energy consumption of layer $\xi$ can be calculated as

$$\text{Power}(\xi) = 0.9\text{pJ} \times \text{SOPs}(\xi) \tag{16}$$

where $0.9$pJ is the energy consumption per synaptic operation (SOP) [Indiveri et al., 2015, Hu et al., 2018, Zhou et al., 2022]. The number of synaptic operations at the layer $\xi$ of an SNN is estimated as

$$\text{SOPs}(\xi) = T \times \gamma \times \text{FLOPs}(\xi) \tag{17}$$

where $T$ is the number of times step required in the simulation, $\gamma$ is the firing rate of input spike train of the layer $\xi$, and $\text{FLOPs}(\xi)$ is the estimated floating point operations at the layer $\xi$.

For classical artificial neural networks, the theoretical energy consumption required by the layer $\xi$ can be estimated by

$$\text{Power}(\xi) = 4.6\text{pJ} * \text{FLOPs}(\xi) \tag{18}$$

Note that $1\text{J} = 10^3 \text{ mJ} = 10^{12} \text{ pJ}$.

## D  Discussion of Limitations

In the image classification task, spiking neural networks have demonstrated comparable performance to ViT on CIFAR-10-DVS and DVS-128-Gesture datasets, which are neuromorphic event-based image datasets created using dynamic vision sensors. We think that the performance gap bewteen SNNs and ANNs in language tasks is mainly due to the lack of neuromorphic language datasets. It is unfair to evaluate SNNs on the datasets that were created to train and evaluate ANNs because these datasets are mostly processed by continuous values. However, it is quite hard to convert language to neuromorphic information without information loss. We hope there will be a new technology to transfer senteces to neuromorphic spikes.

In addition, GPU memory poses a limitation in our experiments. Spiking neural networks have an additional dimension, denoted as $T$ (time step), compared to artificial neural networks. Increasing the number of time steps allows for capturing more information but results in an increased demand for GPU memory by a factor of $T$. During our experiments, we observe that maintaining the same number of time steps during training requires reducing the sentence length of input sentences, which significantly constrains the performance of our models. We remain optimistic that future advancements will provide GPUs with sufficient memory to support the functionality of SNNs.

---

[5]`https://www.cs.cornell.edu/people/pabo/movie-review-data/`

[6]`https://raw.githubusercontent.com/SophonPlus/ChineseNlpCorpus/master/datasets/ChnSentiCorp_htl_all/ChnSentiCorp_htl_all.csv`

[7]`https://raw.githubusercontent.com/SophonPlus/ChineseNlpCorpus/master/datasets/waimai_10k/waimai_10k.csv`

