# OpenReview forum: "SpikeBERT: A Language Spikformer Trained with Two-Stage Knowledge Distillation from BERT"
_NeurIPS.cc/2023/Conference — Submitted to NeurIPS 2023_

### Official Review · Reviewer_DsCq · 2023-06-29

**Soundness:** 3 good
**Presentation:** 3 good
**Contribution:** 3 good
**Rating:** 6
**Confidence:** 4

**Summary:**

This paper proposes SpikeBERT, a spiking-based BERT model for text classification. It employs LIF spiking neurons an surrogate gradients for backpropagation. The training method consists of a two-stage distillation process (pre-training + task-specific). The experiments conducted on different benchmarks for English and Chinese languages show that the proposed SpikeBERT achieves higher accuracy than prior spike-based language models and lower energy consumption than the (non-spiking) BERT.

**Strengths:**

1.	The proposed method is novel and relevant to the community.
2.	The technical sections are described clearly.
3.	The experiments provide good-quality results.


**Weaknesses:**

1.	The design decisions are not discussed in detail.
2.	Unlike non-spiking BERT, there are scalability issues when increasing the depth.


**Questions:**

1.	In Section 3, please discuss the design decisions made to devise the proposed method. Why choosing LIF neurons instead of other neuron models? Why these choices for the architectural parameters?
2.	Since the SpikeBERT does not achieve a significant accuracy increase when increasing the depth, can the proposed method scale up and acquire similar properties like the emergent abilities in (non-spiking) large language models?


**Limitations:**

The limitations have not been discussed by the authors, but there are no major limitations in this work.

---

> ### Author Rebuttal · Authors · 2023-08-09
>
> Thank you for your valuable comments.
>
> #### **Q1: Why choosing LIF neurons instead of other neuron models? Why these choices for the architectural parameters?**
>
> R1: Thank you for your reminding.
>
> Firstly, LIF neuron is one of the most widely used spiking neurons, and other neurons can be seen as some variants of it. In works [1][2] on spiking neural networks and natural language processing, the neuron they chose was LIF neuron. Therefore, we just follow them to use LIF neuron.
>
> Secondly, in order to distill knowledge from the teacher model more effectively, we use the same the architectural parameters as the teacher model BERT-base. For example, the hidden dimension is 768 and model depths is 12. However, the sentence length is not 512 due to the limited GPU memory (See Q3).
>
> We have added these explanations in the revised manuscript.
>
> #### **Q2: Can the proposed method scale up and acquire similar properties like the emergent abilities in (non-spiking) large language models?**
>
> R2: Firstly, as discussed in Section 4.5, the reason why the accuracy of SpikeBERT is generally insensitive to the model depths is that the gradients error may accumulate with the increase of model depths due to the surrogate gradients. What’s more, the depths we used on the experiments were only 8, 12 and 18. Should you wish to explore the potential performance fluctuations across a broader range of depths, we are readily prepared to conduct additional experiments at your behest.
>
>  Secondly, the emergent ability of large language models was proposed by [3]. Large language models usually refer to large **generative** language models, which are mostly decoder-only and mainly used for text generation, such as ChatGPT. SpikeBERT is an encoder-only language **representation** model for language understanding tasks, which is similar to BERT. However, SpikeBERT can be easily extended to the decoder-only models because they are all Transformer-based.
>
> #### **Q3: It seems the limitations have not been discussed by the authors.**
>
> R3: Thank you very much for strong recognition of our work. Our Limitations section is not within the main body of the manuscript, but rather in **Appendix D** (Page 15 of the PDF file). In this section, we primarily discuss the following limitations:
>
> Firstly, there are many neuromorphic event-based image datasets, such as CIFAR-10-DVS and DVS-128-Gesture, which perfectly align with the characteristics of SNN networks. However, such datasets are lacking in the natural language processing tasks.
>
> Secondly, the data used for SNN training introduces an additional temporal dimension (T dimension) compared to traditional data. Limited to the GPU memories, we had to reduce the sentence length of input sentences, which significantly constrains the performance of our models.
>
> ##### Reference:
>
> [1] Lv C, Xu J, Zheng X. Spiking Convolutional Neural Networks for Text Classification[C]. // The Eleventh International Conference on Learning Representations. 2022.
>
> [2] Zhu R J, Zhao Q, Eshraghian J K. Spikegpt: Generative pre-trained language model with spiking neural networks[J]. arXiv preprint arXiv:2302.13939, 2023.
>
> [3] Wei J, Tay Y, Bommasani R, et al. Emergent abilities of large language models[J]. arXiv preprint arXiv:2206.07682, 2022.

---

> > ### Comment · Reviewer_DsCq · 2023-08-14
> > **Response to rebuttal**
> >
> > I would like to thank the authors for answering the reviewers' questions.
> >
> > In light of all the reviewers' comments and author responses, my score is confirmed.

---

### Official Review · Reviewer_9eKU · 2023-07-06

**Soundness:** 3 good
**Presentation:** 3 good
**Contribution:** 3 good
**Rating:** 6
**Confidence:** 3

**Summary:**

This paper presents SpikeBERT, an improved version of the Spikformer spiking transformer model for language tasks. SpikeBERT utilizes a two-stage knowledge distillation method that combines pre-training with BERT and fine-tuning with task-specific data. Experimental results show that SpikeBERT outperforms state-of-the-art spiking neural networks and achieves comparable results to BERT on text classification tasks for English and Chinese, while consuming significantly less energy.

**Strengths:**

1 The writing is commendable.
2 The proposed approach is captivating and innovative, and I believe it will generate significant interest within the machine learning community.
3 The results are promising, particularly in achieving comparable performance to BERT in text classification.

**Weaknesses:**

1 Although SpikeBERT significantly reduces energy consumption during inference, the two-stage knowledge distillation process may introduce additional costs. It would be helpful if the authors could provide insights on this matter.

2 Considering other model architectures such as the OPT model families, does SpikeBERT possess the potential to be extended to these models? It would be interesting to explore the applicability of SpikeBERT beyond the BERT architecture.

3 I am curious about the evaluations of SpikeBERT on additional tasks such as GLUE, RACE, and SQuAD. It would provide valuable insights into the model's performance across a broader range of language processing tasks.

**Questions:**

See Weaknesses

**Limitations:**

No limitations

---

> ### Author Rebuttal · Authors · 2023-08-09
>
> Thank you for your valuable comments.
>
>
>
> #### **Q1: Although SpikeBERT significantly reduces energy consumption during inference, the two-stage knowledge distillation process may introduce additional costs. It would be helpful if the authors could provide insights on this matter.**
>
> R1: Yes, the energy consumption is mainly reduced at the inference time. Once spiking neural networks (SNNs) are well software-trained, they can be deployed on neuromorphic hardware for energy-efficient computing. However, mature on-chip training solutions are not yet available, and it remains a great challenge due to the lack of efficient training algorithms, even in a software training environment. Thank you for pointing it out, and we have made it clear in the revised version.
>
>
>
> #### **Q2: Does SpikeBERT possess the potential to be extended to the models like OPT?**
>
> R2: Yes, SpikeBERT can be easily extended to the generative language models like OPT. As shown in Figure 1 (b), we have improved Spikformer in its architecture, making it possible to process languages. Our SpikeBERT is an encoder-only language **representation** model for language understanding tasks. Therefore, we can easily extend it to decoder-only generative language models by (1) replacing the spiking self-attention module (bi-direction) with spiking **mask** self-attention module (uni-direction), and (2) utilizing the pre-training strategies and corpus of GPT[2][3][4]. Actucally, we have started the research on language generation of SNNs and we are also interested in training large language models (LLMs) based on SNNs.
>
>
>
> #### **Q3: The evaluations of SpikeBERT on additional tasks such as GLUE, RACE, and SQuAD.**
>
> R3: Thank you for your good suggestion.
>
> (1) The selection of the datasets on Table 1 is based on the fact that previous SNNs-related work[5] also used these datasets, which facilitates meaningful comparisons.
>
> (2) Actually, we reported the performance of SpikeBERT on GLUE benchmark in our original manuscript, but we found that the performance of our baseline, SNN-TextCNN[5], on GLUE was extremely poor, and even failed to converge on some tasks, so we think that such a comparison is unfair.
>
> (3) The performances of the baseline model and SpikeBERT on GLUE benchmark are shown in the following table:
>
> |                | SST2  | MRPC  | RTE   | QNLI  | MNLI-(m/mm) | QQP   | CoLA                  | STS-B                  |
> | -------------- | ----- | ----- | ----- | ----- | ----------- | ----- | --------------------- | ---------------------- |
> | Metric         | Acc   | F1    | Acc   | Acc   | acc         | F1    | Matthew’s correlation | Spearman’s correlation |
> | FT BERT        | 92.31 | 89.80 | 69.31 | 90.70 | 83.82/83.41 | 90.51 | 60.00                 | 89.41                  |
> | SNN-TextCNN[5] | 80.91 | 80.62 | 47.29 | 56.23 | 64.91/63.69 | 0.00  | -5.28                 | 0.00                   |
> | SpikeBERT      | 85.39 | 81.98 | 57.47 | 66.37 | 71.42/70.95 | 68.17 | 16.86                 | 18.73                  |
>
> We find that the performance of the Natural Language Inference (NLI) task (QQP, QNLI, RTE) is not satisfactory. The possible reason is that we mainly focus on the semantic representation of a single sentence in the pre-training distillation stage. In the future, we intend to explore the incorporation of novel pre-training loss functions to enhance the model's ability to model sentence entailment effectively.
>
>
>
> Reference:
>
> [1] Zhou Z, Zhu Y, He C, et al. Spikformer: When spiking neural network meets transformer[J]. arXiv preprint arXiv:2209.15425, 2022.
>
> [2] Radford A, Narasimhan K, Salimans T, et al. Improving language understanding by generative pre-training[J]. 2018.
>
> [3] Radford A, Wu J, Child R, et al. Language models are unsupervised multitask learners[J]. OpenAI blog, 2019, 1(8): 9.
>
> [4] Brown T, Mann B, Ryder N, et al. Language models are few-shot learners[J]. Advances in neural information processing systems, 2020, 33: 1877-1901.
>
> [5] Lv C, Xu J, Zheng X. Spiking Convolutional Neural Networks for Text Classification[C]. // The Eleventh International Conference on Learning Representations. 2022.

---

> > ### Comment · Reviewer_9eKU · 2023-08-18
> > **Thank you**
> >
> > I would like to thank the authors for their rebuttal. I will maintain my original rating.

---

### Official Review · Reviewer_XMQ4 · 2023-07-07

**Soundness:** 2 fair
**Presentation:** 3 good
**Contribution:** 2 fair
**Rating:** 3
**Confidence:** 3

**Summary:**

The paper presents SpikeBERT, an implementation of BERT-based models on a Spiking Neural Network (SNN) architecture, motivated by theoretical energy efficiency benefits.

The paper presents the transformer architecture and a two-stage distillation approach which first distills a general purpose BERT model into a general purpose SpikeBERT, then finetunes the SpikeBERT by distilling a finetuned BERT model.

The approach is evaluated on several text classification tasks on English and Chinese datasets, resulting in accuracies lower but comparable than a standard finetuned BERT classifier.

A theoretical energy efficiency improvement is calculated, reporting improvements, which however I find dubious (see weaknesses)

**Strengths:**

- Interesting work on an unusual architecture

**Weaknesses:**

I am especially concerned about the claims of improved energy efficiency, which serve as the main motivation of the paper.
Starting from the introduction, where the author claim: "However, it requires too much computational power and energy to train and deploy state-of-the-art ANN models, leading to a consistent increase of energy consumption per model over the past decade. The energy consumption of large language models, such as ChatGPT[OpenAI, 2022] and GPT-4[OpenAI, 2023], is unfathomable even during inference."
It is clearly not true that "it requires too much computational power and energy to train and deploy state-of-the-art ANN models" since these models are in fact trained and deployed.

More concerning is the theoretical energy comparison of SpikeBERT and BERT (Section 4.4 and Appendix C), where the authors compare FLOPs for BERT and SOPs (spiking operations) for SpikeBERT, multiply by theoretical energy costs and declare SpikeBERT the winner. The theoretical energy costs seem to be copied from other papers, and following the citation chain they seem to come from Yao et al. 2022 "Attention Spiking Neural Networks" where they are computed using data from Horowitz 2014 "1.1 computing’s energy problem (and what we
can do about it)", with the assumption of 32-bit floating point operations on 45nm hardware. Modern GPUs use 7nm hardware, and inference is often done with 8-bit floating point operation or less, therefore I wonder whether these number are obsolete.


**Questions:**

Table 2: mJ is a measure of energy, not power

**Limitations:**

Yes

---

> ### Author Rebuttal · Authors · 2023-08-10
>
> #### **Q1: The claims of improved energy efficiency.**
>
> R1: We apologize for any confusion this may have caused.
>
> Our claim of improved energy efficiency is that the energy consumption is mainly reduced **at the inference time**. Once spiking neural networks (SNNs) are well software-trained, they can be deployed on neuromorphic hardware for energy-efficient computing. However, mature on-chip training solutions are not yet available, and it remains a great challenge due to the lack of efficient training algorithms, even in a software training environment. Thank you for pointing it out, and we have made it clear in the revised version.
>
>
>
> #### **Q2: The energy calculation equation is obsolete.**
>
> R2: Thank you for your suggestion. We have revised the content the Appendix C based on "Attention Spiking Neural Networks"[1] and have incorporated the results of energy consumption calculations based on the new standards you suggested:
>
> $
> E=E\_{MAC}\times\mathrm{FL}\_{\text{SNN Conv}}^{1}+E\_{AC}\times(\sum\_{n=2}\^N\text{SOP}\_{\text{SNN Conv}} \^ n + \sum \_ { m = 1 }\^M\text{SOP}\_{\text{SNN FC}} \^ m + \sum \_ { l = 1 }\^L\text{SOP}\_{\text{SSA}}\^l)
> $
> , where $E\_{MAC} = 4.6pJ$ and $E\_{AC} = 0.9pJ$.
>
> The original energy consumption figures have also been retained for reference.
>
>
>
> #### **Q3: mJ is a measure of energy, not power**
>
> R3: Thank you for your reminding. After conducting a thorough investigation, we have found your observation to be highly accurate. Errors similar to the one you pointed out exist in the works we have followed, such as [2] and [3]. We will rectify this issue in the revised manuscript.
>
>
>
> Reference:
>
> [1]Yao M, Zhao G, Zhang H, et al. Attention spiking neural networks[J]. IEEE transactions on pattern analysis and machine intelligence, 2023.
>
> [2]Zhou Z, Zhu Y, He C, et al. Spikformer: When spiking neural network meets transformer[J]. arXiv preprint arXiv:2209.15425, 2022.
>
> [3]Lv C, Xu J, Zheng X. Spiking Convolutional Neural Networks for Text Classification[C]. // The Eleventh International Conference on Learning Representations. 2022.

---

> > ### Author Response · Authors · 2023-08-15
> > **Theoretical Energy Consumption**
> >
> > According to [1][2], for spiking neural networks (SNNs), the theoretical energy consumption of layer $l$ can be calculated as: $Energy(l) = E_{AC}\times SOPs(l)$, where SOPs is the number of spike-based accumulate (AC) operations.
> > For classical artificial neural networks, the theoretical energy consumption required by the layer $b$ can be estimated by $Energy(b) = E_{MAC}\times FLOPs(b)$, where FLOPs is the floating point operations of $b$, which is the number of multiply-and-accumulate (MAC) operations.
> > We assume that the MAC and AC operations are implemented on the 45nm hardware [3], where $E_{MAC} = 4.6pJ
> > $ and $E_{AC} = 0.9pJ$.
> >
> > The number of synaptic operations at the layer $ξ$ of an SNN is estimated as $SOPs(ξ) = T × γ × FLOPs(ξ)$, where $T$ is the number of times step required in the simulation, $γ$ is the firing rate of input spike train of the layer $ξ$.
> >
> > Therefore, we estimate the theoretical energy consumption of SpikeBERT as follows:
> >
> > $
> > E_{SpikeBERT} =E_{MAC}\times\mathrm{EMB}_{\text{Emb}}^{1} + E\_{AC}\times\\left(\sum\_{m = 1}\^M\text{SOP}\_{\text{SNN FC}}^m+\sum\_{l=1}^L\text{SOP}\_{\text{SSA}}^l\\right)
> > $
> >
> > where $\mathrm{EMB}\_{\text{Emb}}^{1}$ is the embedding layer of SpikeBERT.
> > Then the SOPs of $n$ SNN Fully Connected Layer (FC) and $l$ SSA are added together and multiplied by $E_{AC}$.
> >
> > The energy consumption per sample of fine-tuned BERT and SpikeBERT during inference on 6 text classification benchmarks is as follows:
> >
> > | Dataset  | Model     | Parameters(M) | FLOPs/SOPs(G) | Energy(mJ) | Energy  Reduction | Accuracy(%) |
> > | -------- | --------- | ------------- | ------------- | ---------- | ----------------- | -------- |
> > | ChnSenti | FT BERT   | 109           | 22.46         | 103.38     |                   | 89.48    |
> > |          | SpikeBERT | 109           | 28.47         | 30.51      | 70.49%↓           | 86.36    |
> > | Waimai   | FT BERT   | 109           | 22.46         | 103.38     |                   | 90.27    |
> > |          | SpikeBERT | 109           | 27.81         | 29.90      | 71.08%↓           | 89.66    |
> > | MR       | FT BERT   | 109           | 22.23         | 102.24     |                   | 87.63    |
> > |          | SpikeBERT | 109           | 26.94         | 28.03      | 72.58%↓           | 80.69    |
> > | SST-2    | FT BERT   | 109           | 22.23         | 102.24     |                   | 92.31    |
> > |          | SpikeBERT | 109           | 27.46         | 28.54      | 72.09%↓           | 85.39    |
> > | Subj     | FT BERT   | 109           | 22.23         | 102.24     |                   | 95.90    |
> > |          | SpikeBERT | 109           | 25.92         | 26.96      | 73.63%↓           | 93.00    |
> > | SST-5    | FT BERT   | 109           | 22.23         | 102.24     |                   | 50.41    |
> > |          | SpikeBERT | 109           | 26.01         | 27.33      | 73.27%↓           | 46.11    |
> >
> > We will rewrite Table 2 and Appendix C in the revised manuscript.
> >
> > Reference:
> >
> > [1]Yao M, Zhao G, Zhang H, et al. Attention spiking neural networks[J]. IEEE transactions on pattern analysis and machine intelligence, 2023.
> >
> > [2]Zhou Z, Zhu Y, He C, et al. Spikformer: When spiking neural network meets transformer[J]. arXiv preprint arXiv:2209.15425, 2022.
> >
> > [3]Horowitz M. 1.1 computing's energy problem (and what we can do about it)[C]//2014 IEEE international solid-state circuits conference digest of technical papers (ISSCC). IEEE, 2014: 10-14.

---

> > > ### Author Response · Authors · 2023-08-20
> > > **We look forward to hearing your insightful thoughts!**
> > >
> > > Dear reviewer XMQ4:
> > >
> > > We greatly appreciate your time and effort in reviewing our work. We have carefully considered your suggestions and made the necessary revisions.
> > > Specifically, we have incorporated your advice by modifying the energy consumption formula as per your recommendation. Additionally, we have recalculated the data presented in Table 2. We believe these updates address the concerns you raised in your previous review.
> > > Please see our Official Comments named **Theoretical Energy Consumption** for more details.
> > > Your expertise and insights are highly valued, and we are eager to ensure that our research meets the highest standards.
> > > We are looking forward to hearing more about your insightful thoughts, and would be happy to answer more follow-up questions.

---

### Official Review · Reviewer_PVbb · 2023-07-14

**Soundness:** 3 good
**Presentation:** 2 fair
**Contribution:** 2 fair
**Rating:** 3
**Confidence:** 4

**Summary:**

This work develops SpikeBERT, which extends Spikformer to perform language processing tasks, and proposes a two-stage knowledge distillation method for better training it. Experiments validate the improved accuracy of SpikeBERT over previous SNNs and the improved efficiency over vanilla BERT.

**Strengths:**

1. This work is the first transformer-based SNNs for language processing tasks.

2. Experiments validate the achieved efficiency improvement over vanilla BERT.

**Weaknesses:**

I have the following concerns about this work:

1. The novelty and technical contributions of this work are limited: the modifications from Spikformer to SpikeBERT are minor, and similar distillation schemes have been widely studied and adopted in efficient BERT works, e.g., TinyBERT, TernaryBERT, FastBERT, DistilBERT, etc. It is hard to tell the key technical contribution of this work.

2. The experimental validation is insufficient: It only validates the improved efficiency over vanilla BERT while the aforementioned efficient BERT variants are not benchmarked, making it not clear whether SpikeBERT is a practical efficient BERT option. In addition, the theoretical power is not enough for indicating the real-device efficiency and on-device measurement is highly desirable for benchmarking the aforementioned efficient BERT variants.

3. This work may violate the formatting regulations, i.e., it adds an extra appendix in the main manuscript. In addition, the citation format seems to not follow the official template.

**Questions:**

I have listed my questions in the weakness section.

**Limitations:**

This work may violate the formatting regulations, i.e., it adds an extra appendix in the main manuscript.

---

> ### Author Rebuttal · Authors · 2023-08-10
>
> Thank you for your valuable comments.
>
>
>
> #### **Q1: The novelty and technical contributions of this work are limited.**
>
> R1: (1) Model Architecture: As shown in Figure 1, in addition to introducing the word embedding layer and the layer normal module, we also make the shape of the attention map yielded by Spiking Self Attention (SSA) to be N × N, rather than D × D, where D and N denote dimensionality of hidden layers and the length of inputs respectively. This is crucial.
>
> (2) Knowledge Distillation: As discussed in the Introduction section, deep spiking neural networks (SNNs) directly trained with backpropagation through time (BPTT) using surrogate gradients could suffer from the problem of gradient vanishing or exploding due to “self-accumulating dynamics”. Meanwhile, through pre-experiments, we found that directly training deep SNNs language model is difficult to converge, but they can learn better representation by distillation method. Therefore, we choose knowledge distillation as our training method.
>
> Additionally, this work is the first transformer-based SNNs for language processing tasks and is among the first to show the feasibility of transferring the knowledge of BERT-like language models to spiking-based architectures that can achieve comparable results but with much less energy consumption.
>
>
>
> #### **Q2: The experimental validation is insufficient. The aforementioned efficient BERT variants are not benchmarked.**
>
> R2: Efficient BERT variants like TinyBERT[1], DistilBERT[2], FastBERT[3] reduce energy consumption by reducing the number of model parameters during inference. As the model parameters decrease, FLOPs decrease as well.
>
> However, our SpikeBERT has the same number of model parameters as BERT. Spiking neural networks (SNNs) do not lead to a reduction in the number of model parameters; however, they do effectively decrease synaptic operations (SOPs) so that they can reduce the energy consumption as mentioned in Appendix C. Once SNNs are well software-trained, they can be deployed on **neuromorphic hardware** for energy-efficient computing.
>
> If we increase the number of model parameters for TinyBERT or DistilBERT to the same, then they do not lead to a reduction in energy consumption.
>
> Thank you for your valuable reminding. We will add this explanation to the revised manuscript.
>
>
>
> #### **Q3: This work may violate the formatting regulations**
>
> R3: (1) Regarding the extra appendix in the main manuscript, we apologize for any confusion this may have caused. This was not intentional and can be attributed to an oversight during the final editing process. We assure you that we will promptly rectify this issue by removing the extra appendix and ensuring that the document adheres to the formatting regulations in the revised manuscript.
>
> (2) After conducting a thorough investigation, we find that the citation format we used is a common one, which is also used by [4][5] etc. NeurIPS does not provide us a with a uniform citation format. For example, [6] uses number like “[1][2][3]” to cite reference paper in the manuscript, while [4][5] uses author name and year like “(Ramesh et al., 2022)” to cite. We will change our citation format as you want in the revised version.
>
>
>
>
>
> Reference:
>
> [1] Jiao X, Yin Y, Shang L, et al. Tinybert: Distilling bert for natural language understanding[J]. arXiv preprint arXiv:1909.10351, 2019.
>
> [2] Sanh V, Debut L, Chaumond J, et al. DistilBERT, a distilled version of BERT: smaller, faster, cheaper and lighter[J]. arXiv preprint arXiv:1910.01108, 2019.
>
> [3] Liu W, Zhou P, Zhao Z, et al. Fastbert: a self-distilling bert with adaptive inference time[J]. arXiv preprint arXiv:2004.02178, 2020.
>
> [4] Wang L, Zhou Y, Wang Y, et al. Regularized molecular conformation fields[J]. Advances in Neural Information Processing Systems, 2022, 35: 18929-18941.
>
> [5] Artemev A, An Y, Roeder T, et al. Memory safe computations with XLA compiler[J]. Advances in Neural Information Processing Systems, 2022, 35: 18970-18982.
>
> [6] Liu F, Yang B, You C, et al. Retrieve, reason, and refine: Generating accurate and faithful patient instructions[J]. Advances in Neural Information Processing Systems, 2022, 35: 18864-18877.

---

> > ### Comment · Reviewer_PVbb · 2023-08-15
> > **Reviewer response**
> >
> > Thank the author for their efforts in providing the rebuttal. However, my concerns regarding the novelty and the benchmark with efficient BERT variants are not well solved.
> >
> > In particular, regarding the novelty, the size of the attention map in commonly adopted transformers is typically NxN and thus the proposed modification is intuitive; and knowledge distillation is also a widely adopted (almost "default") setting for model compression.
> >
> > When it comes to benchmarking against efficient BERT variants, the authors' explanation did not address my concern. This is because it remains unclear what rationale they have for benchmarking energy consumption under the same number of parameters, rather than evaluating the overall energy-accuracy trade-off. The authors are expected to benchmark this trade-off to justify whether SpikeBERT is the way to go as compared to previous efficient BERTs.
> >
> > I tend to hold my score for now and I'm willing to discuss with other reviewers to further adjust my score.

---

> > > ### Author Response · Authors · 2023-08-18
> > >
> > > We appreciate your insightful comments and the opportunity to provide further clarification:
> > >
> > > **1. Novelty**:
> > >
> > > (1) In previous research[1], only a simple TextCNN network was employed for single-sentence text classification experiments, which demonstrated the feasibility of using Spiking Neural Networks (SNNs) in Natural Language Processing (NLP) tasks. Our work, however, elevates the application of SNNs in NLP to a new level. Specifically, we have successfully implemented large-scale pretraining via knowledge distillation and achieved state-of-the-art (SOTA) results on a single-sentence classification dataset. This represents a significant advancement over the initial proof-of-concept provided in the earlier study. This work is the first Transformer-based SNNs for language processing tasks and we believe that this field will continue to evolve and advance.
> > >
> > > (2) While it may seem like a natural idea to use knowledge distillation after direct training fails, applying it to spiking neural networks is still poses significant challenges. Firstly, how to align the spiking signals of the student model with the floating-point signals of the teacher model? We addressed this issue by introducing an external “MLP+LayerNorm” layer to convert the spiking signals. Secondly, training spiking neural networks typically requires specific training techniques to stabilize and accelerate the convergence process, so the traditional knowledge distillation methods may not adapt well to these techniques, resulting in training difficulties or suboptimal performance. We employed many training tricks, some of which were not explicitly mentioned in the paper. These tricks included dynamically adjusting the alignment signal weight ratios based on loss ratios, selectively ignoring representations from certain layers, and using longer warm-up periods. In practice, achieving a convergent spiking neural network language model is challenging because traditional knowledge distillation methods and SNNs training methods are ineffective in these scenarios.
> > >
> > > **2. Benchmarking against efficient BERT variants**:
> > >
> > > (1) We conducted experiments on the SST-2 dataset to compare our SpikeBERT with other BERT variants:
> > >
> > > | Dataset | Model         | **Parameters(M)** | FLOPs/SOPs(G) | Energy(mJ) | Energy  Reduction | Accuracy(%) |
> > > | ------- | ------------- | ----------------- | ------------- | ---------- | ----------------- | ----------- |
> > > | SST-2   | FT BERT       | 109.0             | 22.23         | 102.24     | -                 | 92.31       |
> > > |         | SpikeBERT     | 109.0             | 27.46         | 28.54      | 72.09%↓           | 85.39       |
> > > |         | TinyBERT[2]   | 67.0              | 11.30         | 52.01      | 49.13%↓           | 91.60       |
> > > |         | DistilBERT[3] | 52.2              | 7.60          | 34.98      | 65.78%↓           | 90.40       |
> > >
> > > The results of energy consumption calculations is based on the new standards Reviewer XMQ4 suggested.
> > >
> > > (2) We think that the energy efficiency achieved by spiking neural networks (SNNs) is distinct from methods such as knowledge distillation or model pruning, which aim to reduce model parameters. They represent **different technological pathways**. Spiking neural networks do not alter the model parameters but instead **introduce temporal signals** to enable the model to operate in a more biologically plausible manner on neuromorphic hardware. The energy reduction of spiking neural networks is still an estimate, and future advancements in hardware are expected to further decrease energy consumption while potentially accelerating inference speeds [4]. This represents a promising avenue for implementation on artificial intelligence. In the future, we aim to reduce the firing rate of spikes in the network while maintaining accuracy, or achieve higher performance while maintaining similar energy consumption levels.
> > >
> > > We hope that these explanations address your concerns effectively, and we look forward to hearing your insightful thoughts! Thank you!
> > >
> > >
> > >
> > > Reference:
> > >
> > > [1]Lv C, Xu J, Zheng X. Spiking Convolutional Neural Networks for Text Classification[C]. // The Eleventh International Conference on Learning Representations. 2022.
> > >
> > > [2] Jiao X, Yin Y, Shang L, et al. Tinybert: Distilling bert for natural language understanding[J]. arXiv preprint arXiv:1909.10351, 2019.
> > >
> > > [3] Sanh V, Debut L, Chaumond J, et al. DistilBERT, a distilled version of BERT: smaller, faster, cheaper and lighter[J]. arXiv preprint arXiv:1910.01108, 2019.
> > >
> > > [4]Horowitz M. 1.1 computing's energy problem (and what we can do about it)[C]. // 2014 IEEE international solid-state circuits conference digest of technical papers (ISSCC). IEEE, 2014: 10-14.

---

### Official Review · Reviewer_nA4G · 2023-07-24

**Soundness:** 2 fair
**Presentation:** 3 good
**Contribution:** 1 poor
**Rating:** 4
**Confidence:** 3

**Summary:**

The authors have proposed SpikeBERT which is an energy efficient Spiking Neural Networks(SNN) for natural language representation. The architectural design for SpikeBERT is inspired from Spikformer which is an SNN for computer vision, with the following major changes:
	- Spiking Patch Splitting for images is replaced by embedding layer to encode words/tokens into vectors.
	- BatchNorm replaced by LayerNorm
	- Shape of Spiking Self Attention is changed to adapt to language tasks by changing its size based on length on input instead of dimensionality of hidden layers.

The model is trained using a two-step knowledge distillation approach:
	- General purpose: The model is trained using embedding and intermediate hidden layer representations of BERT on unlabeled natural language texts.
Task specific: The model is tuned using task specific logits from a fine-tuned BERT model.

**Strengths:**

	- The paper is well written and easy to follow. The authors provide the necessary background on SNNs including the advantages and challenges in training them.
	- The motivation is clear and the energy consumption presented in the results section helps convey the same.
	- The metrics indicate that the proposed approach outperforms the previous spiking network based baseline developed for natural language understanding and standard SNN training mechanism using surrogate gradients.
	- The authors present a thorough ablation analysis for all the contributions presented in the paper.


**Weaknesses:**

	- The novelty of the paper is limited:
		○ The architecture is mostly derived from Spikformer: the usage of word embeddings instead of image patches, and using layer normalization in transformer  is a standard approach.
		○ The two-step knowledge distillation approach has been used widely in the past for distilling BERT and GPT style transformer models to smaller/specific architectures.
		○ The usage of hidden layer representations in the distillation process is also a standard practice for BERT style models.
		○ Most of the formulations and approaches related to spiking neurons, its derivatives and feature transformations are adapted from previous work.


**Questions:**

	- BERT was introduced several years ago, and since then, multiple adaptions of its architecture and training tasks have been proposed that outperform it such as RoBERTa. Why have the authors chosen BERT as a teacher model for SpikeBERT?


**Limitations:**

	- The datasets used for evaluation seem limited. With models like BERT, it's a standard approach to present results on all GLUE tasks or at least the ones such as MNLI as they are considered to be good & reliable indicators of model quality.

---

> ### Author Rebuttal · Authors · 2023-08-09
>
> Thank you for your valuable comments.
>
> #### **Q1: The novelty (4 aspects) of the paper is limited.**
>
> R1: (1) As shown in Figure 1, in addition to introducing the word embedding layer and the layer normal module, we also make the shape of the attention map yielded by Spiking Self Attention (SSA) to be N × N, rather than D × D, where D and N denote dimensionality of hidden layers and the length of inputs respectively. This is crucial.
>
> (2) As discussed in the Introduction section, deep spiking neural networks (SNNs) directly trained with backpropagation through time (BPTT) using surrogate gradients could suffer from the problem of gradient vanishing or exploding due to “self-accumulating dynamics”. Meanwhile, through pre-experiments, we found that directly training deep SNNs language model is difficult to converge, but they can learn better representation by distillation method, which is consistent with [1]. Therefore, we choose knowledge distillation as our training method.
>
> (3) Although the usage of hidden features is a standard practice of knowledge, the alignment of hidden features between ANNs and SNNs is quite difficult in our work. As discussed in Section 3.3.1, due to the teacher model’s features being in floating-point format, while the student model’s features are integers and involve an additional dimension T (time step), we need external modules to align the features (See Equation 6). Even so, we still find it hard to align the features generated by the student model with those generated by BERT for the first few layers.
>
> (4) We apologize for the lack of precision in our formulas. I find that most of the papers [1][2][3] related to knowledge distillation have similar formulas, but I can rewrite our formulas in the revised manuscript as you want.
>
> Additionally, this work is the first transformer-based SNNs for language processing tasks and is among the first to show the feasibility of transferring the knowledge of BERT-like language models to spiking-based architectures that can achieve comparable results but with much less energy consumption.
>
> #### **Q2: Why have the authors chosen BERT as a teacher model for SpikeBERT?**
>
> R2: Our approach is applicable to any model similar to BERT and RoBERTa. Existing literature[4] indicates that BERT and RoBERTa exhibit minimal substantive differences in downstream tasks, and both models share a similar network architecture. The divergence between these models is primarily observed in the aspect of masking strategy, input tokenization, and training strategy [5], yet these differences do not impact the conclusions drawn in this paper. Moreover, as BERT serves as a representative example of large pre-trained models, we chose BERT as the teacher model.
>
>
>
> #### **Q3: The datasets used for evaluation seem limited. The evaluations of SpikeBERT on GLUE.**
>
> R3: Thank you for your good suggestion.
>
> (1) The selection of the datasets on Table 1 is based on the fact that previous SNNs-related work[6] also used these datasets, which facilitates meaningful comparisons.
>
> (2) Actually, we reported the performance of SpikeBERT on GLUE benchmark in our original manuscript, but we found that the performance of our baseline, SNN-TextCNN[6], on GLUE was extremely poor, and even failed to converge on some tasks, so we think that such a comparison is unfair.
>
> (3) The performances of the baseline model and SpikeBERT on GLUE benchmark are shown in the following table:
>
>
>
> We find that the performance of the Natural Language Inference (NLI) task (QQP, QNLI, RTE) is not satisfactory. The possible reason is that we mainly focus on the semantic representation of a single sentence in the pre-training distillation stage. In the future, we intend to explore the incorporation of novel pre-training loss functions to enhance the model's ability to model sentence entailment effectively.
>
> |                | SST2  | MRPC  | RTE   | QNLI  | MNLI-(m/mm) | QQP   | CoLA                  | STS-B                  |
> | -------------- | ----- | ----- | ----- | ----- | ----------- | ----- | --------------------- | ---------------------- |
> | Metric         | Acc   | F1    | Acc   | Acc   | acc         | F1    | Matthew’s correlation | Spearman’s correlation |
> | FT BERT        | 92.31 | 89.80 | 69.31 | 90.70 | 83.82/83.41 | 90.51 | 60.00                 | 89.41                  |
> | SNN-TextCNN[6] | 80.91 | 80.62 | 47.29 | 56.23 | 64.91/63.69 | 0.00  | -5.28                 | 0.00                   |
> | SpikeBERT      | 85.39 | 81.98 | 57.47 | 66.37 | 71.42/70.95 | 68.17 | 16.86                 | 18.73                  |
>
>
>
> Reference:
>
> [1]Qiu H, Ning M, Yuan L, et al. Self-Architectural Knowledge Distillation for Spiking Neural Networks[J]. 2022.
>
> [2]Tang R, Lu Y, Liu L, et al. Distilling task-specific knowledge from bert into simple neural networks[J]. arXiv preprint arXiv:1903.12136, 2019.
>
> [3]Jiao X, Yin Y, Shang L, et al. Tinybert: Distilling bert for natural language understanding[J]. arXiv preprint arXiv:1909.10351, 2019.
>
> [4]Qiu X, Sun T, Xu Y, et al. Pre-trained models for natural language processing: A survey[J]. Science China Technological Sciences, 2020, 63(10): 1872-1897.
>
> [5]Liu Y, Ott M, Goyal N, et al. Roberta: A robustly optimized bert pretraining approach[J]. arXiv preprint arXiv:1907.11692, 2019.
>
> [6]Lv C, Xu J, Zheng X. Spiking Convolutional Neural Networks for Text Classification[C]. // The Eleventh International Conference on Learning Representations. 2022.

---

> > ### Comment · Reviewer_nA4G · 2023-08-16
> >
> > Thank you, authors, for providing answers to the questions and sharing more data points.
> >
> > Based on the responses, the concerns related to novelty still persist. Specifically, word embeddings, attention matrix quadratic in sequence length and knowledge distillation are well studied and established practices for natural language processing in deep learning and I'm concerned that the current approach mostly combines these existing techniques with Spike Neural Networks.
> >
> > Also, metrics on some large GLUE tasks such as QNLI and QQP seem quite inferior indicating limited applicability of the approach.
> >
> > Therefore, I'll maintain the score as of now.

---

> > > ### Author Response · Authors · 2023-08-17
> > >
> > > We appreciate your insightful comments and the opportunity to provide further clarification:
> > >
> > > (1) In previous research[1], only a simple TextCNN network was employed for single-sentence text classification experiments, which demonstrated the feasibility of using Spiking Neural Networks (SNNs) in Natural Language Processing (NLP) tasks. Our work, however, elevates the application of SNNs in NLP to a new level. Specifically, we have successfully implemented large-scale pretraining via knowledge distillation and achieved state-of-the-art (SOTA) results on a single-sentence classification dataset. This represents a significant advancement over the initial proof-of-concept provided in the earlier study. This work is the first Transformer-based SNNs for language processing tasks and we believe that this field will continue to evolve and advance. Although our current model's performance on the natural language inference (NLI) task is unsatisfactory, we believe that in the future, we can also expand our paradigm to capture the relationships between sentence entailment.
> > >
> > > (2) While it may seem like a natural idea to use knowledge distillation after direct training fails, applying it to spiking neural networks is still poses significant challenges. Firstly, how to align the spiking signals of the student model with the floating-point signals of the teacher model? We addressed this issue by introducing an external “MLP+LayerNorm” layer to convert the spiking signals. Secondly, training spiking neural networks typically requires specific training techniques to stabilize and accelerate the convergence process, so the traditional knowledge distillation methods may not adapt well to these techniques, resulting in training difficulties or suboptimal performance. We addressed this issue by employing many training tricks, some of which were not explicitly mentioned in the paper. These tricks included dynamically adjusting the alignment signal weight ratios based on loss ratios, selectively ignoring representations from certain layers, and using longer warm-up periods. In practice, achieving a convergent spiking neural network language model is challenging because traditional knowledge distillation methods and SNNs training methods are ineffective in these scenarios.
> > >
> > > (3)  As spiking neural networks operate on specialized neuromorphic hardware, energy consumption is expected to decrease further with advancements in neuromorphic hardware technology. Our model has achieved over 70% energy savings under the same parameter settings, while maintaining comparable performance. Furthermore, according to [2], networks running on neuromorphic hardware exhibit faster inference speeds.
> > >
> > > We hope that these explanations address your comments effectively, and we look forward to hearing your insightful thoughts! Thank you!
> > >
> > > Reference:
> > >
> > > [1]Lv C, Xu J, Zheng X. Spiking Convolutional Neural Networks for Text Classification[C]. // The Eleventh International Conference on Learning Representations. 2022.
> > >
> > > [2]Horowitz M. 1.1 computing's energy problem (and what we can do about it)[C]. // 2014 IEEE international solid-state circuits conference digest of technical papers (ISSCC). IEEE, 2014: 10-14.

---

### Author Rebuttal · Authors · 2023-08-10

We thank the reviewers for your insightful comments, which helped us to significantly improve the manuscript. The following major changes have been made in the revised manuscript:

(1) We have added the performance of SpikeBERT and baseline models on GLUE benchmark. However, the performance of baseline model, SNN-TextCNN, on GLUE benchmark is extremely poor, and even failed to converge on some tasks. In the future, we will we intend to enhance the model's ability to model sentence entailment effectively.

(2) We have added some text to explain how the design decisions made to devise the proposed method. Through pre-experiments, we found that directly training deep SNNs language model is difficult to converge, but they can learn better representation by distillation method. Therefore, we choose knowledge distillation as our training method. Our model parameter is the same as BERT.

(3) We have explained our claims of energy efficiency in more detail. In contrast to artificial neural networks (ANNs), the reduction in energy consumption of spiking neural networks (SNNs) is primarily observed at the inference time. Once SNNs are well software-trained, they can be deployed on neuromorphic hardware for energy-efficient computing.

(4) We have conducted a set of experiments to re-calculate the energy consumption based on the new standards Review XMQ4 suggested. Once the experimental results are obtained, we will report them.

(5) All the reviewers’ comments have been addressed in the revised version.

(6) We have revised the paper thoroughly and carefully.

---

### Decision · Program_Chairs · 2023-09-21

**Decision:**

Reject

**Comment:**

This paper aims to demonstrate that spiking neural networks (SNNs) can achieve close to SoTA performance on NLP tasks. Toward this, this paper modifies Spikformer -- a spiking transformer architecture proposed in the vision domain -- and utilizes distillation from BERT in both the pretraining and fine-tuning stages. The resulting model, namely SpikeBERT, achieves much better performance compared to prior SNNs in the literature.

The reviewers had asked for additional experiments on GLUE and explanations about the energy efficiency of SNNs. Both of which were addressed by the authors' responses.

However, other key concerns raised by the reviewers were not satisfactorily resolved during the rebuttal and discussion process. One of the major concerns is about novelty as the paper modifies the Skipformer in ways that are quite standard when it comes to using Transformers in NLP. Also, two-stage distillation (during pretraining and finetuning) is quite common in the NLP. To counter this, the authors have mentioned ways to align dense representations of teacher models with those of spiking signals of SNN, as well as various training tricks/techniques that were needed to realize good performance by SNN. A detailed discussion of these along with ablation results will enhance to contribution of the paper and help address some of the novelty-related questions. In addition, addressing the following issues will improve the quality of a future submission:

1) Inclue a comprehensive discussion of various design choices.
2) Consider including a decoder or encoder-decoder variant of SNNs
3) Improve the pre-training distillation stage to verify if that indeed leads to improved performance on NLI tasks.
4) Include results for **smaller BERT model** with the performance comparable to the SpikeBERT model so that one can compare the energy consumption of BERT with that of SpikeBERT when they have similar performance.